# Assessment of Virtual Reality among University Professors: Influence of the Digital Generation

**Álvaro Antón-Sancho** [1] , **Pablo Fernández-Arias** [2] **and Diego Vergara** [2,*]

1    Department of Mathematics and Experimental Science, Catholic University of Ávila, C/Canteros s/n, 05005 Ávila, Spain; alvaro.anton@ucavila.es

2    Department of Mechanical Engineering, Catholic University of Ávila, C/Canteros s/n, 05005 Ávila, Spain; pablo.fernandezarias@ucavila.es

*    Correspondence: diego.vergara@ucavila.es

**Abstract:** This paper conducts quantitative research on the assessment made by a group of 623 Spanish and Latin American university professors about the use of virtual reality technologies in the classroom and their own digital skills in this respect. The main objective is to analyze the differences that exist in this regard due to the digital generation of the professors (immigrants or digital natives). As an instrument, a survey designed for this purpose was used, the validity of which has been tested in the study. It was found that digital natives say they are more competent in the use of virtual reality and value its technical and didactic aspects more highly, although they also identify more disadvantages in its use than digital immigrants. Differences in responses were found by gender and areas of knowledge of the professors with respect to the opinions expressed. It is suggested that universities design training plans on teaching digital competence and include in them the didactic use of virtual reality technologies in higher education.

**Keywords:** virtual environment; higher education; didactic resource; quantitative analysis

## 1. Introduction

Since 2010, several disruptive technologies related to virtual reality (VR) have emerged [1–3]. VR is a technology that allows the creation of virtual environments, based on the three-dimensional simulation of real environments, in which the user can interact [4]. VR technologies are frequently used to recreate objects or situations favorable to the interests of the professional or researcher of the area of knowledge in which it is being applied. In medicine, for example, it is used to recreate immersive situations appropriate for treating phobias, depression, or anxiety disorders [5–7]; in architecture or engineering it is used to immerse oneself in the projects being designed [8]; or in art, where it is applied to the design of three-dimensional environments to create artistic works [9].

There are multiple applications of VR in the educational and training fields, mainly in vocational training and university higher education. VR is applied in sectors as diverse as the training of mining professionals [10,11], the teaching of endoscopic and orthopedic surgeries [12,13], or to the performance of laboratory practices [14]. Within the latter, it has also been developing a line of research that consists of the analysis of the perception that the different agents of the university community have about the didactic benefits of the use of VR [15]. In this sense, the results reported in the specialized literature are disparate, depending on the area of knowledge, or whether the target population is made up of teachers or students, among other factors. Thus, foreign language learners specifically highlight the reduction of anxiety towards the study of the second language generated using VR techniques [16]. Social science students frequently identify technical difficulties in the use of VR. However, they report positive assessments of its didactic use, especially in the reduction of learning anxiety and the increase in social interaction [17,18]. In contrast, science or engineering students value the effectiveness of VR as a better medium than the

traditional physical environments to represent specific objects of their field of study [19–21]. They also find some drawbacks of VR, such as the technical difficulty involved in its use or the accessibility of VR technologies [22].

Professors' opinions about the didactic use of VR are also relevant because they are the ones who can have a realistic and objective view of the learning achievements that VR techniques allow to reach in comparison with other techniques and of the costs of their implementation. In addition, it is necessary to analyze the digital competence and adaptability of professors to these technologies, given that their didactic use strongly requires a vast knowledge on their part. On the other hand, the description of professors' opinions and the identification of their main concerns or assessments can serve to become aware of the differences that may exist between the vision of the professors and the concerns of the students and thus be more versatile in the design of didactic resources.

Among professors, it is commonly emphasized that VR has many didactic benefits, such as the possibility of designing situations that favor learning to work on abstract concepts in a visual and manipulative way, the increase of motivation and involvement of students in academic activities, and the promotion of inquiry-based learning [23]. In this sense, the opinion of university professors coincides with that of primary school teachers [24]. Professors also particularly value the technical aspects of VR [25,26]. However, they identify some limitations that, in their opinion, VR technologies have. Among them, they highlight the high digital competence required for its use by professors or the high costs of its implementation in universities [25,26].

Previous work has served to identify the gaps that exist between the assessments of professors and their students with respect to the use of VR, which allows them to provide clues to professors to design more effective didactic virtual resources and to suggest training plans for professors with respect to the use of VR or funding channels for universities to incorporate these technologies [27,28].

As indicated above, one of the most relevant factors influencing professors' assessments of VR technology is their level of digital competence. The digital competence of professors is part of a family of competencies of a different nature, the development of which has a decisive impact on teaching: (i) technical; (ii) soft; and (iii) digital [29]. In this classification, technical competencies refer to job-specific skills. Soft skills are transversal abilities, such as those that affect the worker's communicative capacity, critical thinking, social and collaborative skills, and problem-solving ability [30]. Finally, digital competence refers to the knowledge and skills necessary for the safe and critical use of the full range of information and communication digital technologies (ICT) [31]. Following the emergence of the Coronavirus (COVID-19) and the consequent impact on the digital migration of learning environments in education, the concern for digital competence of professors has increased significantly [32,33].

Regarding the degree of development of digital competence, Prensky [34,35] defines two generations, hereafter referred to as digital generations: (i) digital immigrants, consisting of those born before 1980; and (ii) digital natives, consisting of those born after 1980. Digital natives are those who were born and raised in a digitized environment, so that computer media are routine for them. Digital immigrants, on the other hand, have not grown up linked to digital technologies, but have developed their digital competence over years of a continuous process of incorporating ICTs into their routines and their personal and professional lives [36–39].

However, Prensky [34,35] not only defined the different digital generations, but also identified the divergence between digital natives and immigrants as the biggest problem facing education today [40]. Inclusion in one or another digital generation may be conditioned by factors such as geographic location or economic or cultural level. However, Prensky's chronological criterion has been abundantly employed in research to distinguish between digital natives and immigrants [33].

In recent years, the specialized literature has identified different attitudes towards digital technologies that go beyond the differences between digital natives and immigrants [41].

For example, a positive attitude towards the use of technologies facilitates the development of digital competence in older adults [42]. Other variables have been identified, such as technological development, or the policies and services developed by each country or geographical area in terms of sustainable access to technologies [43]. These variables are not limited to the digital generation. In fact, in certain Western European countries such as Italy, France, the United Kingdom, Germany and Spain, these factors have led to young adults making more frequent use and demonstrating more skills in this regard than adolescents [44].

Nevertheless, digital generation has been used as a discriminative variable in numerous studies, including some very recent ones, about the digital skills and attitudes of both students [45–48] and professors [49–51]. Given this context, the general objective of this research is to analyze the opinion that professors have of the didactic use of VR technologies in university classrooms, with special attention to the gaps due to the digital generation of professors. To this end, the view of a target population of professors of different areas of knowledge from Spain, Latin America and the Caribbean has been explored.

The opinions analyzed have been expressed in quantitative terms and refer specifically to the degree of satisfaction regarding technical and didactic aspects of VR, an assessment of the expected future projection of its use in higher education classrooms and its possible disadvantages, and a manifestation of the self-concept regarding the digital competence and use of VR of the study participants. A survey designed for this purpose was used to analyze the professors' assessment of VR and its effectiveness as a resource in the classroom, and significant differences were identified according to the digital generation of the participants in the study. Likewise, within each digital generation, the existence of differences by gender or area of knowledge in the expressed assessments was discussed.

## 2. Materials and Methods

### 2.1. Participants

A total of 623 professors from all areas of knowledge and from universities located in Spain and the following 19 Latin American countries participated in the study: Argentina, Bolivia, Brazil, Chile, Colombia, Costa Rica, Dominican Republic, Ecuador, El Salvador, Guatemala, Honduras, Mexico, Nicaragua, Panama, Paraguay, Peru, Puerto Rico, Uruguay, and Venezuela (Figure 1). The participants were chosen through a non-probabilistic convenience sampling process and all of them responded to a survey about their experience of using VR in learning environments in university classrooms that was sent to them through GoogleForms™. Professors responded to the survey voluntarily, freely, and anonymously. All responses were validated (only those responses in which all questions were answered were considered).

### 2.2. Objectives and Variables

The main objective of this research is to quantitatively analyze the opinions that the participating professors have about the didactic use of VR technologies in university classrooms and the identification of significant differences in these assessments due to the digital generation of the professors (digital native or digital immigrant). Specifically, this study seeks to achieve the following specific objectives: (i) to analyze the opinions of the professors surveyed on the different technical dimensions, employability and didactic effectiveness, future projection, and disadvantages of the use of VR in higher education classrooms; (ii) to identify gaps in the above perceptions by the digital generation to which the surveyed professors belong; and (iii) to study whether there are differences by gender or by area of knowledge between the opinions of the professors of the two digital generations (digital immigrants and digital natives) regarding the evaluations expressed on the use of VR.

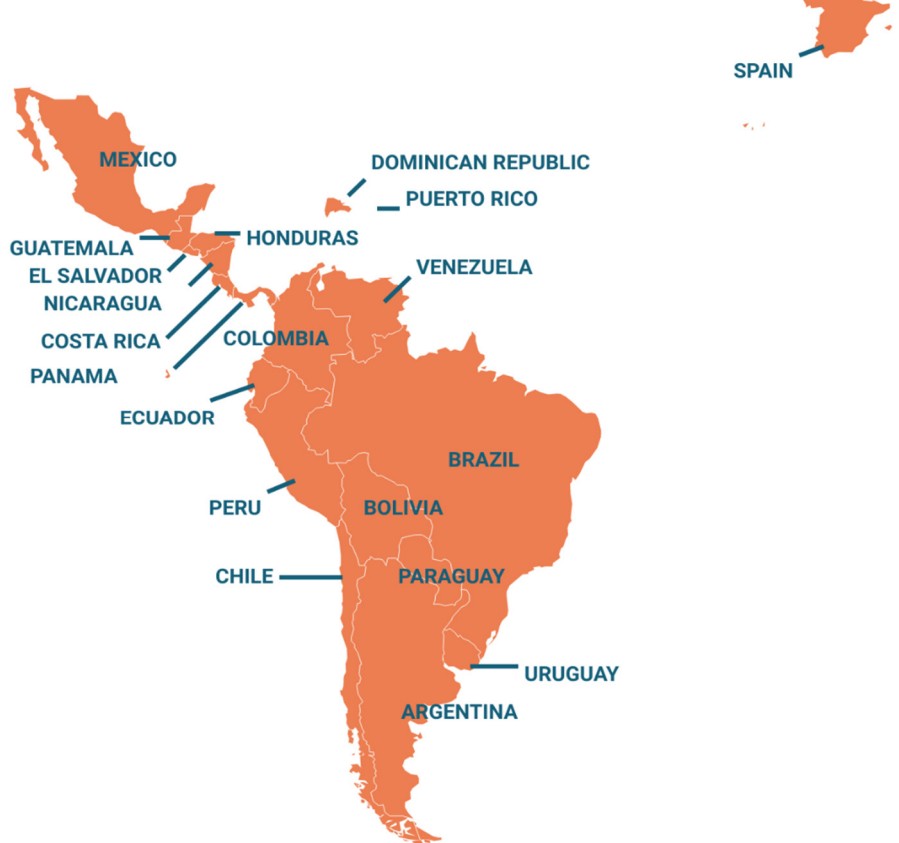

**Figure 1.** Countries participating in the research.

To achieve these objectives, this study distinguishes the following independent variables (Figure 2): (i) digital generation of the participants (dichotomous variable whose values are immigrants and digital natives); (ii) gender (dichotomous); and (iii) participants' area of knowledge (dichotomous). Regarding the latter variable, two major areas have been differentiated, following the International Standard Classification of Education (ISCED), which is the classification established by the United Nations Educational, Scientific and Cultural Organization (UNESCO) [52]: scientific-technical area (composed of the areas of Science, Engineering, Agriculture and Health) and humanistic-social area (composed of the areas of Arts, Humanities, Social Sciences and Law and Services). All independent variables are nominal and dichotomous.

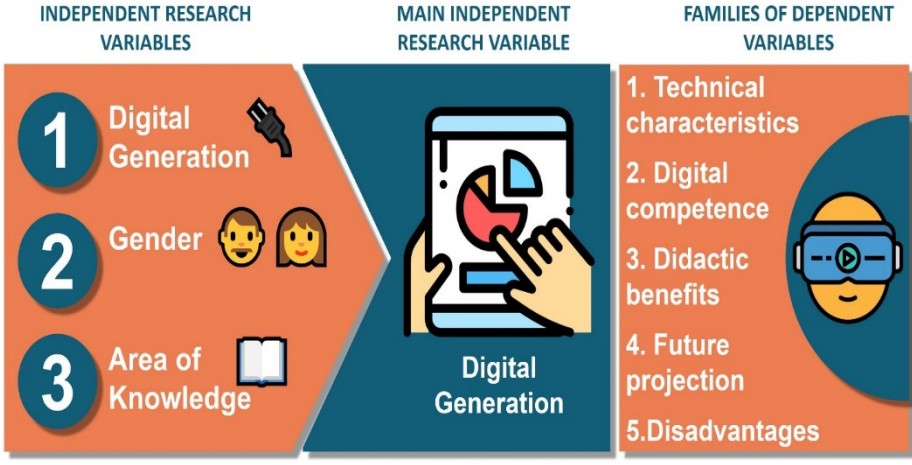

**Figure 2.** Research variables.

The digital generation acts as the main independent variable of the study in the sense that the dependent variables will be analyzed differentiating by digital generations and the gender and area of knowledge will be crossed with the digital generation (i.e., the responses will be differentiated by digital generation within each gender and each area of knowledge). A total of 19 dependent variables, specified in Table 1, were measured which can be grouped into the following families: (i) assessment of the technical characteristics of VR; (ii) self-concept of digital competence and use of VR technologies; (iii) perception of the didactic benefits of using VR in higher education; (iv) future projection that participants perceive about the implementation of VR technologies in university classrooms; and (v) disadvantages of using VR. All the dependent variables are quantitative and will be measured on a scale of 1 to 5, where 1 means the lowest rating and 5 means the highest rating. In the case of the variable measuring the disadvantages of VR, 1 means a minimal perception of the proposed disadvantages and 5 means a high perception of the degree of these disadvantages.

**Table 1.** Questions of the survey.

| Family of Variables | Question Number | Dependent Variable |
|---|---|---|
| Assessment of the following technical aspects of VR | Question 1 | Interaction |
| | Question 2 | User experience |
| | Question 3 | Usability |
| | Question 4 | 3D design |
| | Question 5 | Degree of immersion |
| | Question 6 | Realism |
| Self-concept on the following dimensions of the competence on the use of VR | Question 7 | Digital competence |
| | Question 8 | Training received on VR |
| | Question 9 | VR knowledge |
| Assessment of the following didactic aspects of VR | Question 10 | Academic performance |
| | Question 11 | Student motivation |
| | Question 12 | Course progress |
| VR future forecast | Question 13 | Immersive Virtual Reality (IVR) |
| | Question 14 | Non-Immersive Virtual Reality (NIVR) |
| Disadvantages of VR | Question 15 | Costs |
| | Question 16 | Space required |
| | Question 17 | Scarcity of resources |
| | Question 18 | Lack of knowledge of the professor |
| | Question 19 | Technological obsolescence |

### 2.3. Instrument

In this study, a survey of 19 questions was used, classified into five families. Each question is focused on assessing one of the dependent variables described (Table 1). The survey was designed by the authors for the purposes of this study. All the questions are Likert-type, and their responses are measured on a scale of 1 to 5 (where 1 indicates the lowest valuation and 5 indicates the highest valuation).

### 2.4. Procedure

The research carried out is quantitative and it is based on the participants' responses to the survey described above. The analysis of the distribution of participants according to their digital generation, gender, and area of knowledge was carried out through: (i) goodness of fit test to a homogeneous distribution; (ii) Pearson's test of difference of proportions; and (iii) Fisher's exact test. Once the responses were collected and validated, an Exploratory Factor Analysis (EFA) was applied to the responses to obtain the latent factors that explain the responses obtained. These factors were called subscales of the

survey and correspond to the families of variables described above (Table 1). The theoretical model obtained in the EFA was confirmed by Confirmatory Factor Analysis (CFA). Pearson correlation coefficients of the subscales among themselves and with respect to the overall scale (the whole survey) have been used to obtain the psychometric validation of the instrument. This has been concluded with the analysis of the convergent validity, that has been made through the average variance extracted (AVE), and the analysis of the internal consistency of the instrument through the Cronbach's alpha parameters and composite reliability (CR) of the subscales.

The main descriptive statistics of the responses of the different subscales of the survey were obtained for the analysis of the results. The means and standard deviations of the responses of the subscales were compared when the participants were differentiated by their digital generation using Student's *t*-test and Levene's test, respectively. Finally, the multifactor ANOVA (MANOVA) test was used to compare the mean responses of each subscale when participants were differentiated by their digital generation and by the values of the other independent variables. All mean comparison tests were carried out with Welch's correction without assuming homoscedasticity. It has been taken 0.05 as the level of statistical significance in all tests.

## 3. Results

### 3.1. Distribution of Participants According to the Independent Variables

In the sample of participants there is a slight majority of digital immigrants (59.39%) versus digital natives (40.61%). This is a moderate difference between the frequencies of both generations, but statistically significant considering the goodness-of-fit test statistics to a homogeneous distribution (chi-square = 153.81; *p*-value = 0.0000). From the data shown in Table 2, it is concluded that digital immigrants are more frequent in the sample of participants than digital natives among both males and females, and in both the scientific-technical and humanistic-social areas. The superiority of the frequency of digital immigrants is higher among males and among professors in the scientific-technical area. The Pearson's test statistics for comparison of proportions and the Fisher's exact test confirm that the differences in the frequencies of participants by digital generations within each gender and each area of knowledge are statistically significant.

**Table 2.** Distribution of participants by gender and area of knowledge within each digital generation.

| | | Immigrants (%) | Natives (%) | Pearson's Chi-Square | Pearson's *p*-Value | Fisher's *p*-Value |
|---|---|---|---|---|---|---|
| Gender | Males | 62.2 | 37.6 | 11.865 | 0.0006 * | 0.0006 * |
| | Females | 57.0 | 43.0 | | | |
| Area of knowledge | Scientific-technical | 60.9 | 39.1 | 4.9120 | 0.0267 * | 0.0277 * |
| | Humanistic-social | 57.6 | 42.4 | | | |

\* $p < 0.05$.

### 3.2. Validation of the Survey

An EFA with Varimax rotation on the answers was carried out, which allowed us to identify five latent factors in the survey. Table 3 shows the largest factorial weights for each of the questions in the survey. The five factors identified correspond exactly to the families of variables that define the instrument. From now on, the set of questions forming part of each factor will be called subscale of the survey, so that the term scale will be reserved for the complete survey. The five subscales thus obtained define a theoretical model for the latent factors of observed responses that is statistically significant (chi-square = 3626.50, df = 115, *p*-value = 0.1827). This model explains 64.2% of the variance (Table 4).

The CFA statistics confirm the theoretical model just defined [53]. Indeed, the incremental fit indices are good (AGFI = 0.8885; NFI = 0.8997; TLI = 0.9163; CFI = 0.9287; IFI = 0.9291) and

the absolute fit indices are also appropriate (GFI = 0.9136; RMSEA = 0.0492; AIC = 672.1945; chi-square/df = 3.1743).

**Table 3.** Results of the EFA (Q. means question).

| Question | Factor 1. Technical Aspects of VR | Factor 2. Competence on VR | Factor 3. Didactic Aspects of VR | Factor 4. VR Future Forecast | Factor 5. Disadvantages of VR |
|---|---|---|---|---|---|
| Q. 1 | 0.729 | | | | |
| Q. 2 | 0.612 | | | | |
| Q. 3 | 0.644 | | | | |
| Q. 4 | 0.804 | | | | |
| Q. 5 | 0.640 | | | | |
| Q. 6 | 0.733 | | | | |
| Q. 7 | | 0.849 | | | |
| Q. 8 | | 0.776 | | | |
| Q. 9 | | 0.605 | | | |
| Q. 10 | | | 0.714 | | |
| Q. 11 | | | 0.779 | | |
| Q. 12 | | | 0.796 | | |
| Q. 13 | | | | 0.793 | |
| Q. 14 | | | | 0.740 | |
| Q. 15 | | | | | 0.620 |
| Q. 16 | | | | | 0.763 |
| Q. 17 | | | | | 0.819 |
| Q. 18 | | | | | 0.679 |
| Q. 19 | | | | | 0.614 |

**Table 4.** Cumulative proportion of explained variance of the principal component analysis.

| | Factor 1 Technical Aspects | Factor 2 Competence | Factor 3 Didactic Aspects | Factor 4 Future | Factor 5 Disadvantages |
|---|---|---|---|---|---|
| Proportion Variance | 0.277 | 0.112 | 0.098 | 0.086 | 0.069 |
| Cumulative Variance | 0.277 | 0.389 | 0.487 | 0.573 | 0.642 |

Table 5 shows the Pearson correlation coefficients of the different subscales of the survey, among themselves and with respect to the global scale. These coefficients reveal that there are weak correlations between the different subscales, but moderate correlations between the subscales and the global scale. All coefficients are statistically significant. Indeed, it has been shown that the different subscales are weakly dependent on each other, which implies that each of them is measuring a specific aspect within the study and that none of them is redundant or subsumable in another. The data confirm, therefore, that the model established when defining the different families of variables (Table 2) really explains the responses obtained with statistical significance. This allows checking the psychometric validation of the instrument.

**Table 5.** Pearson correlation coefficients of the subscales among themselves and with respect to the global scale.

| | Technical | Competence | Didactic | Future | Disadvantages | Global |
|---|---|---|---|---|---|---|
| Technical | 1 | 0.1097 | 0.2965 | 0.2918 | 0.1807 | 0.8261 |
| Competence | | 1 | 0.0834 | 0.0611 | 0.0263 | 0.5872 |
| Didactic | | | 1 | 0.3146 | 0.0077 | 0.7936 |
| Future | | | | 1 | 0.1611 | 0.6820 |
| Disadvantages | | | | | 1 | 0.7078 |
| Global | | | | | | 1 |

All Cronbach's alpha and CR parameters are above 0.70 (Table 6), which allows to assume that the instrument used has a high level of internal consistency [54]. In addition, the AVE values exceed 0.50 (Table 6), which also assures convergent validation.

**Table 6.** Cronbach's alpha, CR and AVE parameters of the subscales.

| Subscale | Cronbach's Alpha | CR | AVE |
|---|---|---|---|
| Technical aspects of VR | 0.8842 | 0.8709 | 0.6310 |
| Competence on VR | 0.7590 | 0.7511 | 0.5417 |
| Didactic aspects of VR | 0.8655 | 0.8525 | 0.6184 |
| VR future forecast | 0.8048 | 0.7937 | 0.5972 |
| Disadvantages of VR | 0.8015 | 0.7918 | 0.5926 |

### 3.3. Analysis of the Answers

The participants gave very good assessments to the technical and didactic aspects of VR, intermediate to its future projection as a didactic resource and to the disadvantages it presents, and poor to their own digital competence and for the use of VR (Table 7). In addition, the lowest dispersion occurs in the subscales of technical and didactic aspects of VR, which proves that the opinions in this respect are more homogeneous in these subscales than in the rest (Table 7).

**Table 7.** Mean values and standard deviations of the answers of the different subscales (out of 5).

| | Mean Values | Standard Deviations |
|---|---|---|
| Technical aspects of VR | 4.12 | 0.94 |
| Competence on VR | 2.74 | 1.18 |
| Didactic aspects of VR | 4.23 | 0.83 |
| VR future forecast | 3.78 | 1.00 |
| Disadvantages of VR | 3.63 | 1.21 |

The high deviations of the VR competence, future projection and disadvantages of VR subscales correspond to high deviations in the responses to each of the questions that are part of the different subscales (Figure 3). In addition, there are statistically significant differences between the mean responses of the different items of each subscale, which also influences the high deviations observed.

Specifically, in the VR competency subscale, there is a significant difference between the poor valuations given by the participants to the training received and their knowledge of VR, versus a better valuation of their digital competency (chi-square = 102.70, $p$-value = 0.0000). Participants perceive, in general, that the future projection of NIVR as a teaching resource is lower than that of IVR (chi-square = 30.670, $p$-value = 0.0000). Finally, among the disadvantages of VR, participants highlight its costs and the technical knowledge required for its use against the other proposed disadvantages (chi-square = 63.73, $p$-value = 0.0000).

The digital generation of the participants is statistically discriminative for all the subscales of the survey, except for the subscale on the projection of future implementation of VR (Table 8). There are also statistically significant differences between the deviations of the responses in the subscales of assessment of the technical aspects of VR, its future projection, and its disadvantages (Table 9).

Digital natives report greater knowledge and skill with the use of VR than digital immigrants, in homoscedasticity situation. Digital natives also value VR more highly, both in its technical and didactic aspects, than digital immigrants. In both subscales, the dispersion of responses is greater among the latter, although the difference between the standard deviations of the two generations is not statistically significant in the case of the subscale of didactic aspects. Digital natives also identify the disadvantages of VR to a greater extent than digital immigrants, with a smaller dispersion in their responses.

Regarding the projection of the future of VR, there are no significant differences in the mean responses of the two digital generations, although digital natives give more homogeneous responses than digital immigrants.

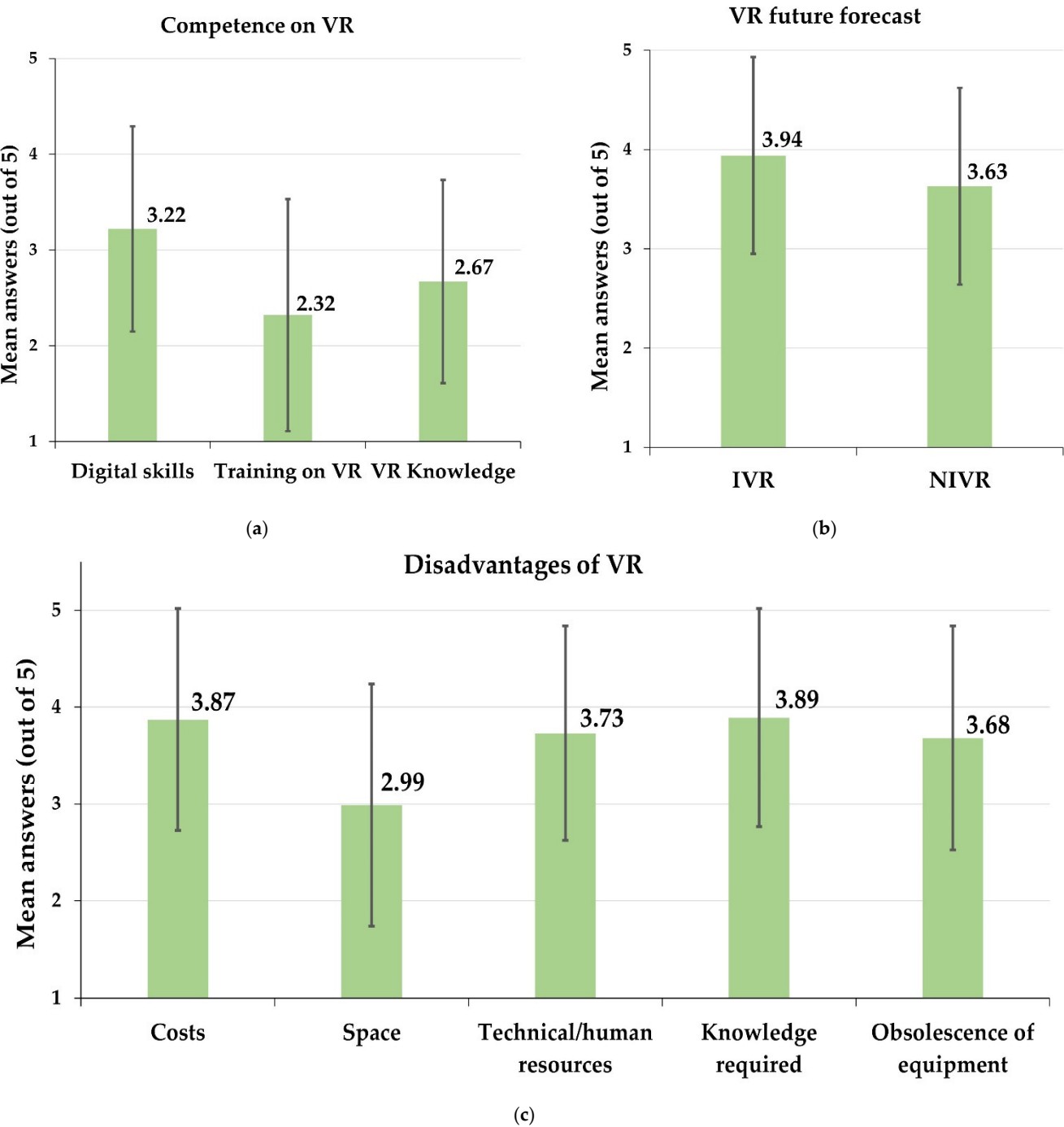

**Figure 3.** Means and deviations of the different questions of the subscales with high standard deviations: (**a**) competence on the use of VR; (**b**) future forecast of VR in higher education; and (**c**) disadvantages of VR.

**Table 8.** Mean values (out of 5) and statistics of the t-Student test when the sample is differentiated by the digital generation.

| | Digital Immigrants | Digital Natives | t | *p*-Value |
|---|---|---|---|---|
| Technical aspects of VR | 4.10 | 4.16 | 4.4131 | 0.0357 * |
| Competence on VR | 2.70 | 2.80 | 3.8829 | 0.0422 * |
| Didactic aspects of VR | 4.21 | 4.26 | 4.8417 | 0.0175 * |
| VR future forecast | 3.79 | 3.78 | 0.0062 | 0.9370 |
| Disadvantages of VR | 3.57 | 3.72 | 12.128 | 0.0005 * |

* $p < 0.05$.

**Table 9.** Standard deviations (over 5) and statistics of the Levene's test when the sample is differentiated by the digital generation.

| | Digital Immigrants | Digital Natives | Levene's F | *p*-Value |
|---|---|---|---|---|
| Technical aspects of VR | 0.94 | 0.92 | 4.4020 | 0.0360 * |
| Competence on VR | 1.15 | 1.21 | 1.3561 | 0.2444 |
| Didactic aspects of VR | 0.85 | 0.80 | 0.7919 | 0.3736 |
| VR future forecast | 1.04 | 0.95 | 4.3148 | 0.0379 * |
| Disadvantages of VR | 1.24 | 1.15 | 8.9765 | 0.0028 * |

* $p < 0.05$.

Gender is a discriminating variable in the subscales for assessing the technical aspects of VR and its future projection, responses are distributed differently between digital generations depending on whether males or females are being considered, as shown by the MANOVA test statistics at the 0.05 level of significance (Table 10). Among males, digital natives value the technical aspects of VR more highly, but among females there is hardly any difference in this respect between digital generations. On the other hand, among females, digital natives have the highest average valuation of the future projection of VR, while among males, it is the digital immigrants who perceive a higher future projection for VR as an educational tool.

**Table 10.** Mean values (over 5) and statistics of the ANOVA test when the sample is differentiated by digital generation and gender.

| | Males | | Females | | MANOVA | *p*-Value |
|---|---|---|---|---|---|---|
| | Immigrants | Natives | Immigrants | Natives | | |
| Technical aspects of VR | 4.06 | 4.20 | 4.13 | 4.12 | 6.5722 | 0.0104 * |
| Competence on VR | 2.75 | 2.81 | 2.65 | 2.78 | 0.5149 | 0.4731 |
| Didactic aspects of VR | 4.19 | 4.31 | 4.22 | 4.22 | 2.4789 | 0.1156 |
| VR future forecast | 3.85 | 3.82 | 3.72 | 3.75 | 4.8102 | 0.0311 * |
| Disadvantages of VR | 3.54 | 3.75 | 3.59 | 3.69 | 1.6029 | 0.2056 |

* $p < 0.05$.

The area of knowledge is a discriminating variable in all subscales except for the future projection of VR (Table 11). Digital natives value the technical and didactic dimensions of VR better than immigrants, but the gap between digital generations is more pronounced in the humanistic-social area in these subscales. However, the gap between generations with respect to competence in the use of VR is wider in the scientific-technical area in favor of digital natives. In both areas of knowledge, digital natives perceive the disadvantages of VR to a greater extent than digital immigrants, but the difference in this respect is notably more pronounced in the humanistic-social area.

**Table 11.** Mean values (over 5) and statistics of the ANOVA test when the sample is differentiated by digital generation and area of knowledge.

|  | Scientific-Technical | | Humanistic-Social | | MANOVA | *p*-Value |
|---|---|---|---|---|---|---|
|  | **Immigrants** | **Natives** | **Immigrants** | **Natives** | | |
| Technical aspects of VR | 4.12 | 4.15 | 4.06 | 4.17 | 6.0228 | 0.0155 * |
| Competence on VR | 2.66 | 2.79 | 2.75 | 2.80 | 6.5873 | 0.0136 * |
| Didactic aspects of VR | 4.24 | 4.29 | 4.16 | 4.22 | 4.5114 | 0.0371 * |
| VR future forecast | 3.83 | 3.77 | 3.73 | 3.79 | 0.8752 | 0.3497 |
| Disadvantages of VR | 3.58 | 3.61 | 3.56 | 3.84 | 7.7943 | 0.0053 * |

\* $p < 0.05$.

## 4. Discussion

The participating professors gave very high valuations to VR as a didactic tool and intermediate-high to its disadvantages and to the possibilities of future implementation in the university. However, they gave lower valuations to their own digital competence and in relation to the use of these technologies (Table 7). Regarding the drawbacks of VR, the costs and the training required to use it stand out (Figure 3). Economic viability is a well-known limitation of the didactic applicability of VR. However, some studies show that the costs of its implementation are decreasing as the technological development increases [55]. Regarding competence in the use of VR, the assessment of the training received on its use is particularly low (Figure 3). These results are in line with other studies which proves that the self-concept of digital competence of professors at different educational levels, areas of knowledge, and geographic location is usually low [56–58]. Typically, the low perception of professors' own digital competence is associated with the lack of experience in the use of digital resources for learning or with the lack of specific training [59]. Although the self-concept of digital competence is not the only variable that should be considered to estimate the digital competence of teachers [60], the low valuations in this regard suggest the need to strengthen faculty training and promote the use of digital teaching technologies by universities.

The good assessments given to VR contrast with the perception of their own digital competence. In fact, as the correlations in Table 5 show, it can be assumed that opinions about the different dimensions of VR are not linearly dependent of the professor's digital competence. On the other hand, there are some correlations that show a certain relationship of linear dependence between some scales, although this relationship is weak. For example, there is a certain dependence between the ratings on didactic aspects of VR and the success of its future implementation with respect to the opinion on its technical features and advantages, although this dependence is weak. However, the good assessments expressed are in line with other studies that analyze the benefits observed by professors who apply VR in the classroom [61]. Some articles study certain specific benefits of VR, such as the experience of reality they generate and their didactic applicability in some specific areas of knowledge, including health sciences [62]. The results obtained support that, within the population studied, digital natives value VR more highly and feel better prepared for its use in the classroom than digital immigrants (Table 8). In addition, digital natives are more confident in their responses because their answers are more homogeneous (Table 9). These results are in line with previous studies, which confirm that digital native professors report higher digital competence than digital immigrants [63–66]. In other studies, although the variable measuring digital generation is not considered, it is also shown that younger professors express greater competence in the didactic use of digital tools [67–69]. However, the digital competence expressed by digital natives is low. This had already been observed in previous studies framed in a population formed by students [45]. Studies analyzing digital competence for the specific use of VR as a teaching resource are scarcer. However, those that exist conclude that, although the teaching effectiveness of VR has been amply proven [70–73], there is an age gap in favor of younger professors [25,74,75]. Other works, which distinguish the assessments by age and teaching experience, show that

it is the experience of using digital environments, and not so much age, the variable that influences the perception of VR [25]. This observation is consistent with the findings of this research because the definition of digital generation involves the experience of using digital technologies and not only a chronological aspect.

This work also brings, as an original perspective, a concretization of the different dimensions of professors' perception of VR. It has been found that the digital generation gap mainly affects the competence for the use of VR, the assessment of its technical and didactic aspects and the identification of the disadvantages of VR (Table 8). Digital natives give higher valuations of VR in all the above respects, but they also identify more of its disadvantages. This may be related to the greater knowledge they express in this regard, which makes them more aware of the limitations of these technologies. This is supported by the results of other previous studies [25]. It has been shown that there are also differences by digital generation and gender in the evaluation of the technical aspects of VR and its future projection in higher education (Table 10). Specifically, among males, it is the digital natives who best value the technical dimensions of VR, while among females, there is hardly any difference between generations. Regarding the future projection of VR, digital natives are slightly more pessimistic among males, but more optimistic among females. These observations allow to deepen the gender differences found in the preceding studies regarding their perception of VR, which are usually focused on a technical area of knowledge [26,76,77]. The results presented here allow to conclude that the digital generation gap is more pronounced among males and suggest the idea that there is a gender gap that affects the use of digital technologies in Latin America, as supported in the preceding literature [78]. However, this last aspect should be studied in greater depth in subsequent studies, given that this work does not have the gender variable as its main object of analysis.

Digital natives give higher valuations of the VR than digital immigrants in the two main areas of knowledge analyzed (scientific-technical and humanistic-social) (Table 11). However, there are no significant differences by area of knowledge in terms of the future perspective of the participants' use of VR, which is high in both areas. In this sense, the results are consistent with those of previous literature, which considers that VR is a didactic resource with a high projection of future implementation [79].

This study opens several lines of future research. A qualitative analysis of the professors' opinions would make it possible to go deeper into the aspects discussed and confirm some of the established theses. In addition, it would be useful to analyze the influence on the opinions expressed of some of the attitudes towards technologies described in the literature [41–44]. In this way, the perspective established here, which takes the digital generation as the main variable, could be complemented. On the other hand, it would be interesting to explore the impact of other sociodemographic variables, such as geographic zone. This would help to analyze, for example, whether the opinions of professors are conditioned by the policies that different countries take on the digitization of society or research investment.

## 5. Conclusions

Participant professors understand, in general, that VR technologies have high technical and didactic potential and are, consequently, highly valued tools in higher education. Although these high perceptions occur irrespective of the digital generation of the professor, digital natives attach higher valuations to VR than digital immigrants. In fact, digital immigrants also identify its disadvantages to a greater extent, probably because of the more realistic view that comes from the fact that they are more experienced in the use of digital technologies, but this hypothesis is not derived from the results of the study and should be confirmed in a subsequent analysis.

The assessments of VR given by participants of each digital generation behave differently depending on the area of knowledge to which they belong (scientific-technical or humanistic-social). In this sense, the difference between the assessments of the participants

of the two digital generations is greater in the humanistic-social area, with the digital natives giving higher assessments. There is also a certain difference in the assessments of VR of each digital generation depending on the gender of the participants. Specifically, among males, digital natives value more the technical aspects of VR but less its future projection, while, among females, it is the digital immigrants who value the technical aspects more and the future projection less.

**Author Contributions:** Conceptualization, Á.A.-S., P.F.-A. and D.V.; methodology, Á.A.-S., P.F.-A. and D.V.; validation, Á.A.-S., P.F.-A. and D.V.; formal analysis, Á.A.-S.; data curation, Á.A.-S., P.F.-A. and D.V.; writing—original draft preparation, Á.A.-S., P.F.-A. and D.V.; writing—review and editing, Á.A.-S., P.F.-A. and D.V.; supervision, Á.A.-S., P.F.-A. and D.V. All authors have read and agreed to the published version of the manuscript.

**Funding:** This research received no external funding.

**Institutional Review Board Statement:** The protocol was approved by the Ethics Committee of the Project "Influence of COVID-19 on teaching: digitization of laboratory practices at UCAV" (13 December 2021).

**Informed Consent Statement:** All participants were informed about the anonymous nature of their participation, why the research is being conducted, how their data will be used and that under no circumstances would their data be used to identify them. The protocol was approved by the Ethics Committee of the Project "Influence of COVID-19 on teaching: digitization of laboratory practices at UCAV" (13 December 2021).

**Data Availability Statement:** The data are not publicly available because they are part of a larger project involving more researchers. If you have any questions, please ask the contact author.

**Conflicts of Interest:** The authors declare no conflict of interest.

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
