# Peer review of "Assessment of Virtual Reality among University Professors: Influence of the Digital Generation"

_computers, doi:10.3390/computers11060092_

Round 1

Reviewer 1 Report

The purpose of this article is to analyse the perceptions of university professors about the use of VR technologies in the classroom taking into account the digital generation gap (digital natives/immigrants) in the use of technology. To this end, the authors utilise a survey among more than 600 professors from universities located in Spain and Latin
American Countries. The subject is interesting due to the extended potential applications of VR in university classroom environments and the data analysis is sound and well described, although the conclusions are somewhat limited.
In view of the above, I recommend this article for publication.

Author Response

Dear reviewer:

The authors would like to thank the reviewer for the attention he/she has shown to our article and for the care he/she has taken with us in carrying out this review. We are very pleased and satisfied that he/she liked it and we appreciate his/her assessment.

Reviewer 2 Report

The paper is, in general, written well. However there are some shortcomings which may detract from the general importance of the paper. It seems the paper places emphasis on the digital natives aspect - citing Prensky's work which was published 2 decades ago. I believe there is more recent research work about attitudes towards technologies that can also be referenced and which would add value to the paper. The scientific method for data collection and analysis is sound and the methodology is also very clear. The soundness of the results is not questionable. However the discussion of the results, lacks some depth and therefore can be improved. If there were additional qualitative data added to the results this would have given more depth to the discussion. 

Author Response

Please, find enclosed a detailed response.

Reviewer 3 Report

A study aimed at quantifying the assessments made by a total of 623 Spanish or Latin American University Professors about the use of virtual reality technologies in the classroom and their own digital skills is described in this paper. The main objective was to look for differences between digital generations, (divided here into "digital immigrants" or "digital natives"). The data collected are presented in correlation tables, and it is concluded that differences between the assessments of participants of the two digital generations appear greater in the human and social sciences field (which is unsurprising), and there seems to be a "gender gap". In the conclusions, the authors recommend that universities offer faculty training courses on what they call "digital competence" for the use of VR technologies in the classroom.

The article needs to undergo major revision before it may be recommended for publication:

1. The study goal is not clear. Why we need insight about what teachers think of the impact of Virtual Reality for the future needs to be scientifically justified. The authors use the word "perception" a lot, but what they measure are coarse opinions collected through google sheets.

2. The experimental design variables are not specified in terms of a clearly understandeable design plan. The Figure provided to that effect does not convey this information. It is of crucial importance that the experimental design variables are embedded in a clear scientific rationale, allowing to understand the motivation for this particular design, and the choice of dependent variables fed into the statistical analyses.

3. The part relative to data analysis is not clear. What are we supposed to learn from the multiple correlation analyses? Correlations do not imply causality. The data appear to be over-interpreted. There does not seem to be any clear gender effect, or the effect is not made clear enough in the data analysis. Where is the claim of a "gender gap" in the findings here derived from?

4. The different correlations foudn here should be interpreted carefully. The conclusions do not appear warranted under the light of the data. Why should the observed differences call for educational programmes on the use or potential usefulness of Virtual Reality in the future?

5. The manuscript needs to be carefully checked for grammar and style once all the major revisions have been implemented.

Author Response

(The authors gave the same response as above.)

Round 2

Reviewer 3 Report

The authors have successfully clarified most of the critical issues raised in my previous review by making substantial and valuable amendements to their paper. While I still do not see a deeper purpose here, I believe the results make a contribution with respect to potential challenges represented by Virtual Reality and other connected technology for teachers and educators. I am therefore happy to recommend this paper for publication in COMPUTERS.